# The Koala Immune Response to Chlamydial Infection and Vaccine Development—Advancing Our Immunological Understanding

**DOI:** 10.3390/ani11020380

**Published:** 2021-02-03

**Authors:** Bonnie L Quigley, Peter Timms

**Affiliations:** 1Provectus Algae, Noosaville, QLD 4566, Australia; bonnie.quigley2017@gmail.com; 2Genecology Research Centre, University of the Sunshine Coast, Sippy Downs, QLD 4556, Australia

**Keywords:** koalas, vaccines, immunity, *Chlamydia*

## Abstract

**Simple Summary:**

Chlamydia is a major pathogen of the Australian marsupial, the koala (*Phascolarctos cinereus*). One approach to improving this situation is to develop a vaccine. Human Chlamydia research suggests that an effective anti-chlamydial response will involve a balance between a cell-mediated Th1 response and a humoral Th2 responses, involving systemic IgG and mucosal IgA. Characterization of koalas with chlamydial disease suggests that increased expression for similar immunological pathways and monitoring of koalas’ post-vaccination can be successful and subsequently lead to improved vaccines. These findings offer optimism that a chlamydial vaccine for wider distribution to koalas is not far off.

**Abstract:**

*Chlamydia* is a significant pathogen for many species, including the much-loved Australian marsupial, the koala (*Phascolarctos cinereus*). To combat this situation, focused research has gone into the development and refinement of a chlamydial vaccine for koalas. The foundation of this process has involved characterising the immune response of koalas to both natural chlamydial infection as well as vaccination. From parallels in human and mouse research, it is well-established that an effective anti-chlamydial response will involve a balance of cell-mediated Th1 responses involving interferon-gamma (IFN-γ), humoral Th2 responses involving systemic IgG and mucosal IgA, and inflammatory Th17 responses involving interleukin 17 (IL-17) and neutrophils. Characterisation of koalas with chlamydial disease has shown increased expression within all three of these major immunological pathways and monitoring of koalas’ post-vaccination has detected further enhancements to these key pathways. These findings offer optimism that a chlamydial vaccine for wider distribution to koalas is not far off. Recent advances in marsupial genetic knowledge and general nucleic acid assay technology have moved koala immunological research a step closer to other mammalian research systems. However, koala-specific reagents to directly assay cytokine levels and cell-surface markers are still needed to progress our understanding of koala immunology.

## 1. Chlamydia and Koalas

*Chlamydia* are obligate intracellular bacteria recognised in a wide range of hosts. Traditionally identified and studied in birds, cattle, guinea pigs, sheep and humans [1], continued research has expanded the list of chlamydial hosts to include insects, amphibians, molluscs, arachnids, reptiles, fish, and amoeba, as well as mammals like pigs, goats, deer, cats, bats, possums, and koalas [2]. Chlamydial disease in the koala, *Phascolarctos cinereus*, has been particularly well studied, given the devastating toll it has taken on this iconic Australian marsupial (Figure 1) [3,4]. Chlamydial infection in koalas is dominated by the species *Chlamydia pecorum*. These infections lead to ocular and urogenital/reproductive diseases comparable to *Chlamydia trachomatis* infections in humans, which include keratoconjunctivitis and scarring in the eye leading to blindness and cystitis/nephritis and reproductive cysts in the urogenital and reproductive tracts, respectively, leading to severe pain and infertility [3,4]. In both humans and koalas, vaccination has been identified as the most promising avenue of control for this pathogen [5]. However, despite years of research, no commercial vaccine is available for either host. What research has achieved is a greater understanding of the immune response to chlamydial infection and vaccination, particularly in koalas, setting the stage for future success.

Marsupials occupy a unique branch of the mammalian evolutionary tree, having diverged from their eutherian (placental) relatives ~148 million years ago [6]. Originally believed to have slower and less accentuated immune responses [7,8,9], it is now recognised that the immune system of marsupials is just as intricate and complex as that of their eutherian counterparts [10]. While study into the components and general development of the koala immune system has already spanned several decades of research (reviewed by [11]), the complete koala genome has only recently been sequenced [12]. This has meant that much of the foundational work carried out with koala immune genes and processes were based on concepts extrapolated from more characterised mammalian systems. Koala-specific reagents for experimentation have also been limited, given the non-model organism status of this marsupial. However, despite these limitations, chlamydial vaccine development for koalas has progressed over the last decade to generate research vaccine formulations with very promising efficacies [13,14,15]. Research has also highlighted many similarities in immune responses to *Chlamydia* infection between hosts, allowing for knowledge from one host to guide research in others. This has proven advantageous to the koala in the field of chlamydial immune responses and vaccine development.

## 2. Effective Chlamydial Immune Responses

Chlamydial infection, disease, and vaccine research, often from human or mouse studies, has established a solid framework for what immune responses are necessary to clear and prevent chlamydial infections. Additionally, chlamydial research in non-model systems, such as non-human primates and guinea pigs, deserves recognition for also advancing ocular chlamydial disease understanding and vaccine development [16]. Overall, it has become well-established that a combined cellular and humoral immune response is needed for complete protection from chlamydial infection and disease progression [17,18,19].

### 2.1. Immunogenetics

For any adaptive immune response to be generated, an early key step is the presentation of chlamydial antigens to T cells via the major histocompatibility complex (MHC) or human leukocyte antigen (HLA) system. MHC molecules present antigens from either intracellular threats via class I molecules or externally phagocytosed antigens via class II molecules to T lymphocytes to initiate an adaptive immune response [20]. As an intracellular bacterium, *Chlamydia* has the potential to interact with both MHC classes. In humans, many studies have looked for associations between specific HLA alleles and susceptibility to chlamydial infection or complications [21]. Immunogenetic studies have found links between chlamydial infections/complications and HLA alleles from both classes, with examples including the presence of alleles from HLA class I A and C loci having significantly higher risk of *C. trachomatis* pelvic inflammatory disease [22] and the HLA class II HLA-DQB1*06 allele emerging as a significant risk marker for chlamydia reinfection in African American women [23]. These genetic links to infection outcome highlight that individuals within a population will mount slightly different responses to the same chlamydial infection, an important consideration during vaccine design.

### 2.2. Cell Mediated (Th1) Responses

Traditionally, the desired immune response to chlamydial infections has been identified as a T helper cell type 1 (Th1) cell-mediated response, with interferon gamma (IFN-γ) being the critical cytokine involved in chlamydial clearance [24]. IFN-γ directly affects the survival of *Chlamydia* through several mechanisms, including enhancing the engulfment and elimination of *Chlamydia* by macrophages [25], activating nitric oxide synthase (iNOS) to produce nitric oxide (NO) to inhibit chlamydial replication [26], and limiting both iron and tryptophan availability for *Chlamydia* growth by downregulating the transferrin receptor [27] and inducing indoleamine-2,3-dioxygenase (IDO) to degrade tryptophan [28], respectively. IFN-γ also affects the survival of *Chlamydia* by inducing T cells to differentiate into Th1 cells and inhibiting proliferation of the T helper cell type 2 (Th2) antibody response [29]. Along with a Th1 adaptive immune response involving CD4/CD8 T cells, an effective cellular response against *Chlamydia* also requires the recruitment of innate cells including macrophages, dendritic cells, and natural killer cells to the mucosal site of infection [17,24].

### 2.3. Antibody (Th2) Responses

While it is generally agreed that a Th1 cell-mediated response is necessary for *Chlamydia* control and protection, the distinct Th1/Th2 paradigm of host defence has encountered major challenges due to the reality that most antigens or vaccines (including chlamydial vaccines) induce mixed immune responses comprising of both humoral and cell- mediated effectors [30]. It has been shown that a robust and protective T-cell memory response against *Chlamydia* requires an effective primary antibody response characterised by specific antibody isotypes whose role is to modulate Th1 activation via Fc receptors that facilitate the rapid uptake, processing, and presentation of pathogen-derived antigens for an enhanced T-cell response [30]. Antibody mediated immunity is increasingly being recognised as necessary, with studies showing the appearance of serum antibodies strongly correlating with chlamydial clearance [31]. Specific examples include the presence of IgA within vaginal secretions correlating with chlamydial clearance [32] and the induction of anti-chlamydial IgG2a and IgA post-vaccination leading to a strong IgG2a recall response post-challenge [33]. Additionally, an important role for antibodies is emerging in the secondary memory response [34], with antibody-mediated neutralization and opsonization [35] and antibody-dependent cellular cytotoxicity (ADCC) [36] identified as important chlamydia control mechanisms.

### 2.4. Inflammatory/Neutrophil (Th17) Responses

The lineage of interleukin 17 (IL-17)-producing CD4 T helper (Th17) cells are also emerging as an important component of the anti-chlamydial response. Th17 cells produce pro-inflammatory cytokines, such as IL-17 and tumour necrosis factor alpha (TNF-α), to act on fibroblasts, macrophages, and endothelial and epithelial cells to recruit granulocytes (especially neutrophils) to the site of infection [29]. Once present, neutrophils in particular have been found to play a critical role in the control of *Chlamydia* in the early stages of infection [37].

### 2.5. Coordinated Responses

Finally, although traditional immunology divides immune responses into discreet Th1, Th2, and Th17 categories, several important anti-chlamydial mechanisms require coordinated action from multiple categories. Examples of these interactions can be seen in the critical antibody class switching to IgG2a and IgA induced by the Th1 cytokines, IFN-γ and transforming growth factor beta (TGF-β) [38] and role of anti-chlamydial antibodies in ADCC by both macrophages and neutrophils for chlamydial clearance [36,39]. In addition, the important role played by tissue-specific memory T cells (Trm), which are not easily categorised within the Th1/Th2/Th17 paradigms, is being increasingly recognised [40,41]. Clearly, many diverse and complex processes are needed to control chlamydial infection in the mammalian host (Figure 2).

## 3. Chlamydial Infection, Disease, and Vaccine Responses in Koalas

It has been proposed that a successful *Chlamydia* vaccine for koalas will need to induce both cellular immune responses through up-regulation of IFN-γ and IL-17, as well as humoral immune responses that generate *Chlamydia*-specific plasma IgG and mucosal IgG and IgA responses with neutralizing capabilities [5]. As such, focused effort has gone into characterising these markers, among others, during both natural *C. pecorum* infection and post-*C. pecorum* vaccination in koalas.

### 3.1. Immunogenetics Related to Chlamydia in Koalas

The koala genome contains 23 MHC class I and 23 MHC class II genes and pseudogenes [12]. Examination of the class I genes determined that 11 genes are actively transcribed in the koala, with three genes ubiquitously expressed as classical class Ia genes (Phci-UA, UB and UC) and eight genes with tissue limited expressions as nonclassical class Ib genes (Phci-UD, UE, UF, UG, UH, UI, UJ and UK) [42]. Survey of the classical class I genes have thus far identified 21 UA, 5 UB, and 12 UC alleles for these genes in the koala population [43]. Even though the number of identified koala class I alleles is expected to increase as more koala populations are surveyed, these numbers are still quite small compared to the thousands of HLA class I alleles recognised in humans [44] (Table 1).

Within the MHC class II gene family, four class II subfamilies are recognised, consisting of alpha and beta subunits of DA, DB, DC and DM [12,45,46]. Studies investigating the allelic diversity of class II subunits within DA and DB genes have found that the beta subfamilies (DAB and DBB) contain more allelic diversity than the alpha subfamilies (DAA and DAA), leading to more focus on the beta subfamilies in research studies [45]. As such, the current collection of sequenced koala MHC class II alleles stands at three DAA, 42 DAB, three DBA, 26 DBB, three DCB, and four DMB alleles, with DCA and DMA alleles yet to be characterised [43,47,48].

Several MHC allele associations have been made with koala chlamydial infections and disease progression. Within koalas from the mid-north coast of New South Wales, Australia, a higher proportion of *Chlamydia*-infected koalas were found to carry the DAB*10 allele relative to non-infected koalas [47]. In the same population, koalas possessing the DBB*04 allele had higher levels of *Chlamydia* heat shock protein 60 (c-hsp60) antibody levels than koalas without DBB*04 [47]. Moving further north to examine koalas already infected with *Chlamydia* in southeast Queensland, Australia, DAB*10 and DBB*04 alleles again emerged with significant associations, but to different circumstances [48]. In these infected koalas, both DAB*10 and UC*01:01 alleles were significantly more prevalent in infected koalas that did not progress to clinical disease, while DBB*04 and DCB*03 alleles were significantly more prevalent in infected koalas that did progress to clinical disease [48]. Finally, a modelling study that also looked at southeast Queensland koalas found that knowing the MHC class II DAB and DBB profiles of a koala improved the likelihood of predicting a koala’s chlamydial disease status and that koalas without DBB*03 were more likely to have clinical chlamydial disease [49]. Collectively, these studies suggest that MHC immunogenetics do play some role in the immune response to *Chlamydia* infection and progression to disease in koalas. However, as with most genetic associations, these links are complex and will require extensive study to be understood.

### 3.2. Cell-Mediated Responses to Chlamydia in Koalas

Given the lack of koala-specific reagents to directly measure specific CD4 or CD8 T cell populations, the primary marker for evaluating Th1 responses in koalas has been to follow IFN-γ expression. In natural infection settings, peripheral blood mononuclear cells (PBMCs) from koalas with active chlamydial disease have been found to have higher expression of IFN-γ and TNF-α than koalas with asymptomatic chlamydial infection or no chlamydial infection/disease [50,51,52]. In large chlamydial vaccination studies of wild koalas, vaccination increases the level of IFN-γ detected [13,53,54]. This IFN-γ increase was seen in koalas regardless of whether the vaccine was formulated with *C. pecorum* major outer membrane protein (MOMP) [13,53] or polymorphic member protein (Pmp) [13] as the target chlamydial antigen. These studies indicate that, as in other hosts, IFN-γ production appears to be an important component of the koala anti-chlamydial and vaccine response.

Interestingly, despite the well-recognised importance of IFN-γ, there have also been smaller studies where positive vaccine outcomes were achieved and increases in IFN-γ levels could not be detected in koalas post-vaccination [14,15]. Although sample timing can always be a factor in detecting cytokine expression, the genetics of *C. pecorum* may also inform on the variable IFN-γ detection. As discussed above, a major mechanism of IFN-γ action against *Chlamydia* is the induced depletion of tryptophan [28]. For chlamydial species like *C. trachomatis*, which lack most of the tryptophan biosynthesis operon, growth is severely inhibited when IFN-γ is present [55]. However, *C. pecorum* possesses a nearly complete tryptophan biosynthesis operon [56,57] and is able to overcome IFN-γ mediated tryptophan depletion by utilising alternative precursors to sustain growth [55]. This suggests that IFN-γ mediated tryptophan depletion may not be as effective against *C. pecorum* as it is against other chlamydial species. It should still be acknowledged that IFN-γ affects *Chlamydia* through multiple mechanisms and remains an important cytokine for chlamydial clearance. However, *C. pecorum* control in koalas may require a coordinated response involving several anti-chlamydial mechanisms, with different studies detecting different mechanisms depending on the experimental design.

### 3.3. Antibody Responses to Chlamydia in Koalas

In a traditional Th2 response, IL-4, -6, -10, and -11 are the hallmark cytokines that initiate an antibody response and lead to the release of IL-4, -5, -9, -10, and -13, characteristic of the Th2 phenotype [29]. Within koalas with current chlamydial disease, significantly higher expression of IL-10 is detected in PBMCs compared to koalas with asymptomatic chlamydial infection and no chlamydial infection/disease [51]. This suggests that some level of Th2 response is generated during chlamydial disease in koalas.

In koala vaccination studies, antibody responses are typically measured as either total systemic anti-*C. pecorum* IgG in plasma or mucosal anti-*C. pecorum* IgG or IgA at the ocular or urogenital site. In every chlamydial vaccine trial in koalas where antibodies were measured, there has been a detectable increase in systemic IgG to *C. pecorum* post-vaccination, regardless of the vaccine formulation tested [5]. Characterisation of these systemic *C. pecorum* IgG antibodies has found that, (a) diverse MOMP genotypes, including genotypes not included as antigen in the vaccine, could be recognised [58], (b) vaccination induced a greater epitope recognition compared to natural infection (including to conserved regions of MOMP) [15,59,60] and that (c) vaccination increased the neutralisation effect of the antibodies generated [60]. Given the fact that *C. pecorum* is currently recognised to have 15 MOMP genotypes associated with koalas [4], achieving antibody responses to multiple MOMP genotype has offered promise that vaccination may induce heterologous protection. Focusing in on the mucosal epithelium, as the site of chlamydial infection, anti-*C. pecorum* mucosal IgG and IgA levels have also been found at higher levels post-vaccination [13,14,61]. This is an important parameter to consider in vaccine efficacy, as mucosal antibodies are part of the primary defence against infecting *Chlamydia*. Overall, chlamydial vaccination in koalas appears to generate a robust antibody response that can contribute to other aspects of the overall anti-*Chlamydia* immune response.

### 3.4. Inflammatory/Neutrophil Responses to Chlamydia in Koalas

The general state of tissue inflammation has come to be recognised as the result of Th17 cells producing IL-17, IL-6, IL-22, and TNF-α to activate several cell linages to recruit effector cells like neutrophils to the site of infection [29]. In koalas, when currently chlamydial diseased animals have been examined, significantly higher expression of IL-17A in PBMCs has been observed [52]. This has suggested that there is a role for the Th17 response in chlamydial disease progression and management in koalas.

Examination of IL-17 expression during chlamydial vaccine trials in koalas has also supported a role for this immune response in *C. pecorum* disease and control. Levels of IL-17 expression have shown post-vaccination increases in wild koala chlamydial vaccine trials [13,53,54]. This has led to increased levels of IL-17 expression being strongly associated with decreases in urogenital chlamydial load and less chlamydial disease in these trials [54]. Looking globally, total transcriptome profiling of koala PBMCs one-month post-vaccination has further expanded links to inflammation-related pathways, with significant up-regulation of 26 genes involved in neutrophil degranulation detected [15]. Collectively, these results continue to support the multi-faceted anti-chlamydial immune response observed in other hosts, with aspects of the Th17 response having a place in vaccination responses in koalas (Figure 3).

## 4. Future Directions

Our understanding of the koala immune system has advanced dramatically over the past decade. While the availability of the complete koala genome, as well as the genomes of closely related marsupials, has contributed greatly to this advancement, so has the extensive effort that has been expended characterising immune responses to chlamydial vaccine development in koalas. Studies continue to reveal similarities between koala immune responses and responses measured in other mammalian model systems. Methodologies utilizing nucleic acids (DNA and RNA) have flourished recently in koala immunology research, with the expanded genomic and transcriptomic data that has become available. The next big step forward will need to be the development of koala-specific reagents to investigate cytokines and specific cell-surface markers to the extents that are currently routine for human or mouse immunology research. The journey for an effective *Chlamydia* vaccine continues for both humans and koalas and progress is being made on both fronts. Expanding the repertoire of immunological assays available for koala vaccine research would allow discoveries in this marsupial to contribute back to overall understanding of anti-chlamydial immune responses, to the benefit of all species. That would be a successful outcome for everyone.

## Figures and Tables

**Figure 1 animals-11-00380-f001:**
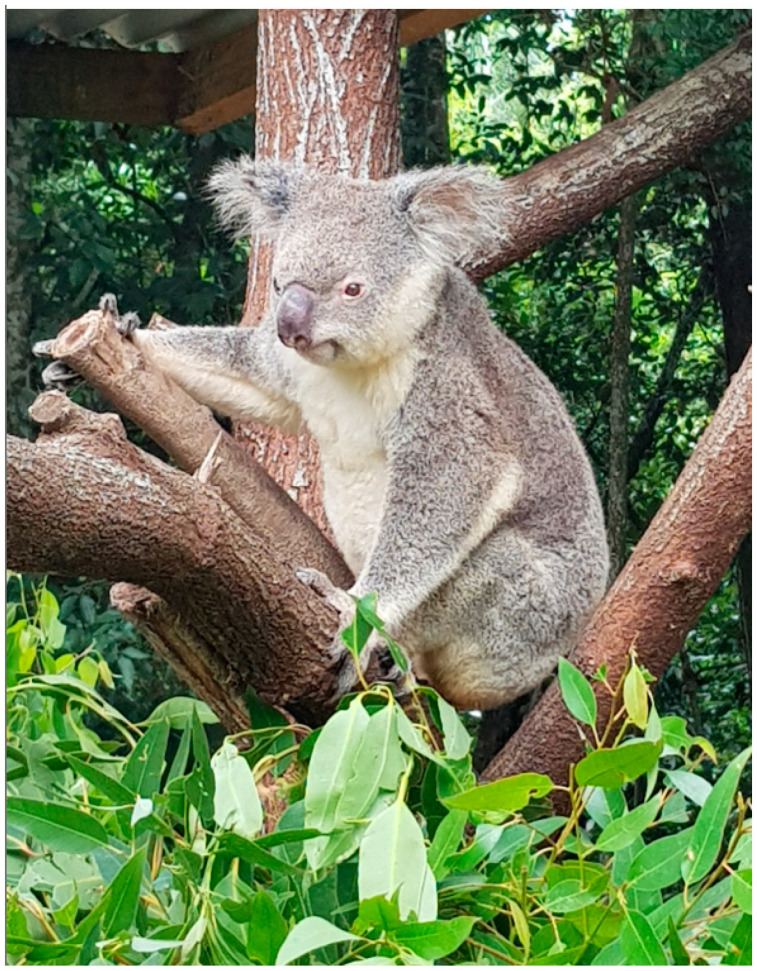
Koala (*Phascolarctos cinereus*). Photo credit Bonnie Quigley.

**Figure 2 animals-11-00380-f002:**
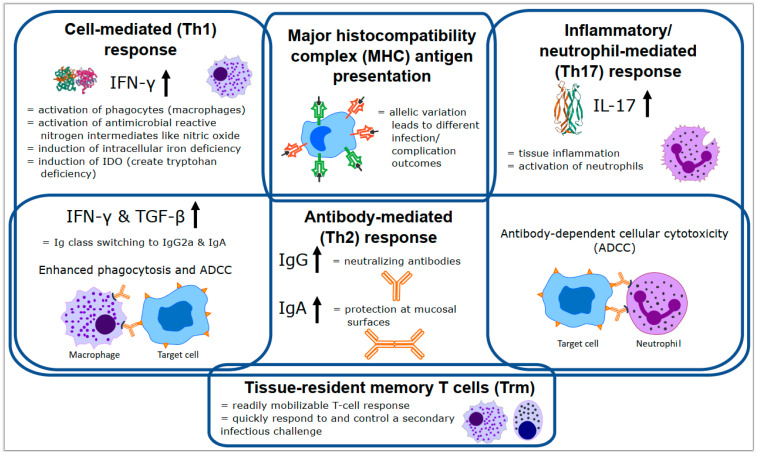
Overview of the major immune responses and components important for controlling chlamydial infection (based on the mouse immune response). Details of each process are described in the text. Th = T helper cell, MHC = major histocompatibility complex, IFN-γ = interferon gamma, IL-17 = interleukin 17, TGF-β = transforming growth factor beta, ADCC = antibody-dependent cellular cytotoxicity, Ig = immunoglobulin, IDO = indoleamine-2,3-dioxygenase.

**Figure 3 animals-11-00380-f003:**
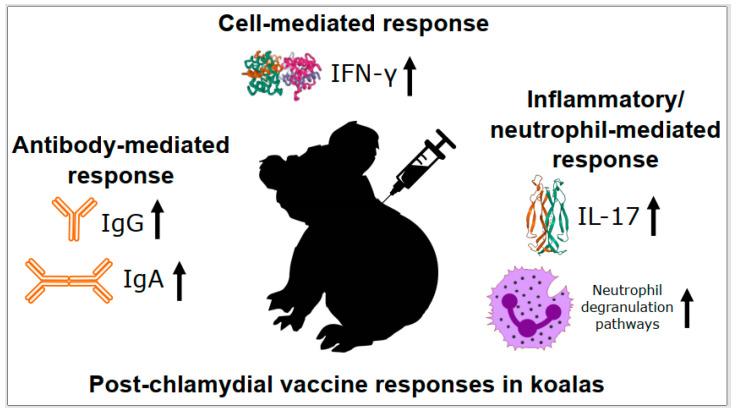
Post-chlamydial vaccine responses in koalas. Details of each process are described in the text. IFN-γ = interferon gamma, IL-17 = interleukin 17, Ig = immunoglobulin.

**Table 1 animals-11-00380-t001:** Major Histocompatibility Complex (MHC)/Human Leukocyte Antigen (HLA) alleles currently recognised in koalas and humans, respectively. Part A—MHC alleles in koalas, Part B—HLA alleles in humans (from [44]).

A	Class	MHC Loci	Number of Alleles	B	Class	HLA Loci	Number of Alleles
	Class I	*UA*	21		Class I	*A*	2480
		*UB*	5			*B*	3221
		*UC*	10			*C*	2196
	Class II	*DAα*	3			*E*	8
		*DAβ*	44			*F*	4
		*DBα*	3			*G*	18
		*DBβ*	26		Class II	*DMα*	4
		*DCα*	No work yet done			*DMβ*	7
		*DCβ*	3			*DOα*	3
		*DMα*	No work yet done			*DOβ*	5
		*DMβ*	4			*DPα1*	22
						*DPβ1*	591
						*DQα1*	34
						*DQβ1*	678
						*DRα*	2
						*DRβ1*	1440
						*DRβ3*	106
						*DRβ4*	42
						*DRβ5*	39

## Data Availability

All data referred to in the manuscript is already published.

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
