# Peer review of "The Koala Immune Response to Chlamydial Infection and Vaccine Development—Advancing Our Immunological Understanding"

_animals, 2021, doi:10.3390/ani11020380_

Round 1

Reviewer 1 Report

The authors presented a well-written review tackling Chlamydia infection in the Australian marsupial, the koala (Phascolarctos cinereus), as well as the immune response, and vaccine prospects.

They made parallels in human and mouse research, stating that it is well-established that an effective anti-chlamydial response will involve a balance of cell-mediated Th1 responses involving interferon-gamma (IFN-γ), humoral Th2 responses involving systemic IgG and mucosal IgA, and inflammatory Th17 responses involving interleukin 17 (IL-17) and neutrophils.

  1. This is, in general, a good parallel, but mice are not suitable either to assess the immune response after ocular chlamydial infection or a protective potential of a vaccine targeting the ocular disease as they fail to develop the ocular disease with C. muridarum or Ct [1]. Koalas are suffering a lot from the ocular infection. Two other animal models: non-human primates (NHPs) and guinea pigs develop the ocular disease when infected with Chlamydiae. Therefore, I would very much appreciate a short paragraph (probably within paragraph 1.1 Effective chlamydial immune responses) about other chlamydial animal models, above all the guinea pig and non-human primate animal models because these two models more closely resemble what we see in koalas. In addition, lots of research about chlamydial vaccines is done using guinea pigs and NHPs (this should be acknowledged). The other important parallel between koalas and guinea pigs is the lack of reagents for both models and this is a huge problem for more targeted studies in different species suffering from the chlamydial infection, which would surely help in the development of an efficient chlamydial vaccine based not only on neutralizing antibodies.
  2. The authors are nicely writing about the humoral and cellular immune response. Nevertheless, there can also be added that Trm cells are also considered protective [2, 3].
  3. Figure 2. Please mark within the text that this figure is based on mouse immune response (BALB/c specifically?) because, in mice, Th1-dependent IFN-γ induces the production of IgG2a (parallel in humans is IgG1).
  4. What do the authors think about prospects to develop koala-specific reagents? Is it realistic that we can have that soon? What about lab-development of PCR-specific reagents for koalas? It is time-consuming but probably the way to go forward.

Happy New Year!

References

  1. Rank RG, Whittum-Hudson JA. Animal models for ocular infections. Methods Enzymol. 1994;235:69-83. Epub 1994/01/01. PubMed PMID: 8057937.
  2. Morrison SG, Morrison RP. In situ analysis of the evolution of the primary immune response in murine Chlamydia trachomatis genital tract infection. Infect Immun. 2000;68(5):2870-9. Epub 2000/04/18. doi: 10.1128/iai.68.5.2870-2879.2000. PubMed PMID: 10768984; PubMed Central PMCID: PMCPMC97499.
  3. Johnson RM, Brunham RC. Tissue-Resident T Cells as the Central Paradigm of Chlamydia Immunity. Infection and Immunity. 2016;84(4):868-73. doi: 10.1128/iai.01378-15.

Author Response

The authors presented a well-written review tackling Chlamydia infection in the Australian marsupial, the koala (Phascolarctos cinereus), as well as the immune response, and vaccine prospects.

They made parallels in human and mouse research, stating that it is well-established that an effective anti-chlamydial response will involve a balance of cell-mediated Th1 responses involving interferon-gamma (IFN-γ), humoral Th2 responses involving systemic IgG and mucosal IgA, and inflammatory Th17 responses involving interleukin 17 (IL-17) and neutrophils.

  1. This is, in general, a good parallel, but mice are not suitable either to assess the immune response after ocular chlamydial infection or a protective potential of a vaccine targeting the ocular disease as they fail to develop the ocular disease with C. muridarum or Ct [1]. Koalas are suffering a lot from the ocular infection. Two other animal models: non-human primates (NHPs) and guinea pigs develop the ocular disease when infected with Chlamydiae. Therefore, I would very much appreciate a short paragraph (probably within paragraph 1.1 Effective chlamydial immune responses) about other chlamydial animal models, above all the guinea pig and non-human primate animal models because these two models more closely resemble what we see in koalas. In addition, lots of research about chlamydial vaccines is done using guinea pigs and NHPs (this should be acknowledged). The other important parallel between koalas and guinea pigs is the lack of reagents for both models and this is a huge problem for more targeted studies in different species suffering from the chlamydial infection, which would surely help in the development of an efficient chlamydial vaccine based not only on neutralizing antibodies.

We thank the reviewer for this good suggestion and have included a statement in the suggested paragraph highlighting the important contribution non-human primates and guinea pigs have made to ocular chlamydial research and vaccine development (along with the reference suggested).

  1. The authors are nicely writing about the humoral and cellular immune response. Nevertheless, there can also be added that Trm cells are also considered protective [2, 3].

Reference to the importance of tissue-resident memory T cells (Trm) has been added to the section “Coordinated responses”, along with the references indicated by the reviewers, Trm has been added to Figure 2.

  1. Figure 2. Please mark within the text that this figure is based on mouse immune response (BALB/c specifically?) because, in mice, Th1-dependent IFN-γ induces the production of IgG2a (parallel in humans is IgG1).

The notation that Figure 2 is based on mouse immune response data has been added to the figure legend.

  1. What do the authors think about prospects to develop koala-specific reagents? Is it realistic that we can have that soon? What about lab-development of PCR-specific reagents for koalas? It is time-consuming but probably the way to go forward.

We agree with the reviewer that koala-specific reagents would be the way forward in this immunology area (which is why we ended the future directions section with this idea). PCR/DNA based reagents have developed with the release of the koala genome, so that is promising. However, other reagents will be a significant challenge to produce simply because the demand is low (limited number of research groups in the field), making the time and money-consuming process of development not profitable for suppliers. If individual research groups can develop enough reagents to share, that might be a way forward. Sadly, that approach will never match the quality or uniformity of a supplier-produced reagent, so we think koala-specific reagents are still probably some time off. 

Happy New Year!

To you, as well ?.

Reviewer 2 Report

Quigley and Timms present a concise overview of mostly their recent work to understand the natural history of infection and immunity to Chlamydia in the koala and many years of work building up to the development and testing of a vaccine. It draws on some selective parallels with human and mouse disease and infection to make an interesting review of interest in comparative immunology and infectious disease. Its concise, well written and presented. I have minor comments only.

Points for clarification

Highlights - given the breath of immune responses discussed wouldn’t it be exceptional if genes in the MHC both polymorphic and non-polymorphic weren’t involved in the controlling the immune response? Unlike the other 3 highlight points which all say something specific I'm not sure this really says anything about the role of MHC in control of specific chlamydial immune responses in natural infection or in response to vaccination in Koalas. The highlight about Th17 response in vaccination could also be more specific for chlamydial disease.

I would prefer to see some further, wider (even if kept brief) and more up to date appreciation of HLA studies for both ocular and chlamydial urogenital disease rather than the citation of a 2009 review article that was selective in its focus on HLA including links to class 2 associations with immune response. I mention this given the importance the manuscript places on MHC in several places including the highlights and work this group has done in typing Koala MHC.

Line 77. -  “In humans, many studies have looked for associations between specific HLA alleles and susceptibility to chlamydial infection or complications (Morré et al., 2009). These immunogenetic studies have found links between chlamydial infections/complications and the presence of different HLA alleles, notably variations within the HLA-DQ family (Morré et al., 2009)”

The sections 1.3, 1.4, 1.5 and 1.6 bring the immune responses types together and are are good, but can you speak to any role of tissue homing cells in one of these sections? And if we are discussing coordinated responses then the contrasting results between candidate gene studies (presented in Morre et al 2009), GWAS and HiHost GWAS type studies for example Yeung et al 2017 Nat Comms and Wang et al PMC6093297.

line 158 three DBB, 26DBB? – Which is correct here 3 DBC or something else. Might be worth considering a diagram of the Koala MHC and its organisation on which chromosome for example just to get to grips with for instance HLA where there are thousands of alleles at some loci. Can anything be drawn comparatively between human hsp60 responses and HLA or with disease and those observed in koalas not the actual alleles but within loci or class?

line 196 - growth

line 200 - does pecorum have a cytotoxin? Is IFNg resisted or growth controlled by alternative pathways analogous muridarum

line 217 - the challenge for human infection has always been lack of heterologous protection. Is it possible to fit in something about polymorphism of pecorum MOMP and whether this presents a problem or not? I understand point a and b in the section below but without recalling the diversity of pecorum MOMP relative to the situation in human hosts its difficult to directly contrast the challenges if we are drawing parallels with human disease.

line 261 - might be useful to cite further examples where vaccine progress is being made for humans in addition to advances in koalas at this point.

Author Response

Quigley and Timms present a concise overview of mostly their recent work to understand the natural history of infection and immunity to Chlamydia in the koala and many years of work building up to the development and testing of a vaccine. It draws on some selective parallels with human and mouse disease and infection to make an interesting review of interest in comparative immunology and infectious disease. Its concise, well written and presented. I have minor comments only.

Points for clarification

Highlights - given the breath of immune responses discussed wouldn’t it be exceptional if genes in the MHC both polymorphic and non-polymorphic weren’t involved in the controlling the immune response? Unlike the other 3 highlight points which all say something specific I'm not sure this really says anything about the role of MHC in control of specific chlamydial immune responses in natural infection or in response to vaccination in Koalas.

We agree with the reviewer that the role MHC plays in control of chlamydia is less well-defined that the other areas of immune research. However, the data being generated does suggest that MHC has a role – so we feel that it is important to highlight this. We feel with more research, a definitive answer to the part MHC plays (or doesn’t play) in chlamydia control will become clearer.

The highlight about Th17 response in vaccination could also be more specific for chlamydial disease.

The highlight has been modified to directly refer to post-vaccination in koalas. It now reads “Increased IL-17 expression post-vaccination has shown positive effects in koalas”.

I would prefer to see some further, wider (even if kept brief) and more up to date appreciation of HLA studies for both ocular and chlamydial urogenital disease rather than the citation of a 2009 review article that was selective in its focus on HLA including links to class 2 associations with immune response. I mention this given the importance the manuscript places on MHC in several places including the highlights and work this group has done in typing Koala MHC.

Line 77. - “In humans, many studies have looked for associations between specific HLA alleles and susceptibility to chlamydial infection or complications (Morré et al., 2009). These immunogenetic studies have found links between chlamydial infections/complications and the presence of different HLA alleles, notably variations within the HLA-DQ family (Morré et al., 2009)”

We have changed the second sentence at Line 77 to indicate more recent and specific examples of HLA allele associations with chlamydial infections/complications. The text at this section now reads: “In humans, many studies have looked for associations between specific HLA alleles and susceptibility to chlamydial infection or complications [21]. Immunogenetic studies have found links between chlamydial infections/complications and HLA alleles from both classes, with examples including the presence of alleles from HLA class I A and C loci having significantly higher risk of C. trachomatis pelvic inflammatory disease [22] and the HLA class II HLA-DQB1*06 allele emerging as a significant risk marker for chlamydia reinfection in African American women [23].”

The sections 1.3, 1.4, 1.5 and 1.6 bring the immune responses types together and are good, but can you speak to any role of tissue homing cells in one of these sections? And if we are discussing coordinated responses then the contrasting results between candidate gene studies (presented in Morre et al 2009), GWAS and HiHost GWAS type studies for example Yeung et al 2017 Nat Comms and Wang et al PMC6093297.

In response to reviewer 1, we have added reference to tissue-resident memory T cells, which is a type of tissue homing cell. For the level of detail we are aiming for in this review, we do not want to delve much deeper into this topic. As for contrasting specific methods between specific studies, we feel that level of detail is beyond the scope for this review. We welcome others to write about these different methods and the results they produce.

line 158 three DBB, 26DBB? – Which is correct here 3 DBC or something else. Might be worth considering a diagram of the Koala MHC and its organisation on which chromosome for example just to get to grips with for instance HLA where there are thousands of alleles at some loci. Can anything be drawn comparatively between human hsp60 responses and HLA or with disease and those observed in koalas not the actual alleles but within loci or class?

Line 158 should have said three DBA, 26 DBB – the typo has been corrected. While we could not envision a diagram to address the suggestion of illustrating the differences between the number of koala MHC and human HLA alleles, we have instead added Table 1 to list the number of known alleles in each species. We feel this table captures the essence of the information reviewer 2 is referring to. Finally, we felt that the request to draw a comparison to human hsp60/HLA to disease verses koala hsp60/MHC to disease is still too premature with the koala data at this time. Hopefully, continued research will generate more evidence of a connection to make this comparison meaningful in the near future.

line 196 – growth

Thank you – the typo “grow” has been corrected to “growth”

line 200 - does pecorum have a cytotoxin? Is IFNg resisted or growth controlled by alternative pathways analogous muridarum

Yes, C. pecorum has 2 cytotoxins (“C. pecorum cytotoxins belonged to two separate gene clusters (Cluster 1:CPE1_0552, CPE2_0552, CPE3_0552; Cluster 2:CPE1_0554, CPE2_0555, CPE3_0555) each showing greatest similarity to cytotoxins from C. muridarum. It is unclear whether the two different cytotoxins in C. pecorum have different biological functions or host specificity. Related cytotoxins in E. coli and C. difficile act by glycosylating small GTP-binding proteins of Rho and Ras families, inhibiting the host signalling and regulatory functions [42], lymphocyte activation [43] and by blocking the induction of IFN-γ.” from Sait M, Livingstone M, Clark EM, et al. Genome sequencing and comparative analysis of three Chlamydia pecorum strains associated with different pathogenic outcomes. BMC Genomics. 2014;15(1):23. Published 2014 Jan 14. doi:10.1186/1471-2164-15-23).  However, to our knowledge, there have not been any studies directly investigating these toxins in C. pecorum, or any other pathways to deal with IFNg other than the tryptophan biosynthesis operon, so we have not mentioned the cytotoxins/other pathways in this review.

line 217 - the challenge for human infection has always been lack of heterologous protection. Is it possible to fit in something about polymorphism of pecorum MOMP and whether this presents a problem or not? I understand point a and b in the section below but without recalling the diversity of pecorum MOMP relative to the situation in human hosts its difficult to directly contrast the challenges if we are drawing parallels with human disease.

We have added a statement at this point indicating the currently recognized polymorphism of C. pecorum MOMP and how the antibody data to date has shown promise for heterologous protection. The statement reads: “Given the fact that C. pecorum is currently recognized to have 15 MOMP genotypes associated with koalas [4], achieving antibody responses to multiple MOMP genotype has offered promise that vaccination may induce heterologous protection.”

line 261 - might be useful to cite further examples where vaccine progress is being made for humans in addition to advances in koalas at this point.

While we appreciate that human chlamydia vaccine progress would be interesting to the readers of this review, that topic is quite extensive and really is the subject of a whole review in itself. Hence, we feel that we would not do the topic justice here and would prefer to leave it out of our current manuscript.
